# Identification and *in-silico* characterization of *taxadien-5α-ol-O-acetyltransferase (TDAT)* gene in *Corylus avellana* L.

**Mona Raeispour Shirazi[1], Sara Alsadat Rahpeyma[1] \*, Sajad Rashidi Monfared[2], Jafar Zolala[1], Azadeh Lohrasbi-Nejad[1]**

1 Department of Agricultural Biotechnology, Shahid Bahonar University of Kerman, Kerman, Iran,
2 Department of Biotechnology, Faculty of Agriculture, Tarbiat Modares University, Tehran, Iran

\* s.rahpeyma@uk.ac.ir

**Data Availability Statement:** All relevant data are within the manuscript and its Supporting Information files.

## Abstract

Paclitaxel® (PC) is one of the most effective and profitable anti-cancer drugs. The most promising sources of this compound are natural materials such as tissue cultures of *Taxus* species and, more recently, hazelnut (*Corylus avellana* L.). A large part of the PC biosynthetic pathway in the yew tree and a few steps in the hazelnut have been identified. Since understanding the biosynthetic pathway of plant-based medicinal metabolites is an effective step toward their development and engineering, this paper aimed to identify *taxadiene-5α-ol-O-acetyltransferase* (*TDAT*) in hazelnut. *TDAT* is one of the key genes involved in the third step of the PC biosynthetic pathway. In this study, the *TDAT* gene was isolated using the nested-PCR method and then characterized. The cotyledon-derived cell mass induced with 150 μM of methyl jasmonate (MeJA) was utilized to isolate RNA and synthesize the first-strand cDNA. The full-length cDNA of *TDAT* is 1423 bp long and contains a 1302 bp ORF encoding 433 amino acids. The phylogenetic analysis of this gene revealed high homology with its ortholog genes in *Quercus suber* and *Juglans regia*. Bioinformatics analyses were used to predict the secondary and tertiary structures of the protein. Due to the lack of signal peptide, protein structure prediction suggested that this protein may operate at the cytoplasm. The homologous superfamily of the T5AT protein, encoded by *TDAT*, has two domains. The highest and lowest hydrophobicity of amino acids were found in proline 142 and lysine 56, respectively. T5AT protein fragment had 24 hydrophobic regions. The tertiary structure of this protein was designed using Modeler software (V.9.20), and its structure was verified based on the results of the Verify3D (89.46%) and ERRAT (90.3061) programs. The T5AT enzyme belongs to the superfamily of the transferase, and the amino acids histidine 164, cysteine 165, leucine 166, histidine 167, and Aspartic acid 168 resided at its active site. More characteristics of *TDAT*, which would aid PC engineering programs and maximize its production in hazelnut, were discussed.

**Funding:** The author(s) received no specific funding for this work.

**Competing interests:** The authors have declared that no competing interests exist.

## Introduction

Cancer is known as a leading cause of death worldwide, and it is expected that the number of cancer patients will increase to over 22 million cases in the next 20 years [1]. Consequently, the demand for anticancer drugs is overgrowing. Paclitaxel (PC), sold under the trade name Taxol, is a chemotherapy medication used to treat various types of cancer such as ovarian and breast cancers as well as AIDS-related Kaposi's Scarcorna [2–4]. Compared to other similar compounds, PC's mode of action against cancer cells is unique in that it inhibits cell proliferation by binding to microtubules. This compound also promotes the stabilization of microtubules at the G2-M phase of the cell cycle [3, 5]. PC is a diterpenoid compound that was first extracted from the western yew tree (*Taxus brevifolia*). Particularly, hazelnut (*Corylus avellana* L.) [6], some microorganisms like yew endophytic fungi, and hazelnut endophytic fungi [7, 8] are the other raw resources for PC production. All natural sources produce low levels of PC [9–11]. Although PC can be synthesized through total chemical synthesis, it is found time-consuming, expensive, and low-yielding due to the complexity of the chemical structure of PC [12, 13]. Hence, in response to the increasing demand for the supplies of PC, new alternative approaches are required to be developed. It has been well investigated that *Taxus* and *C. avellana* L. cell cultures are promising sources for PC production [4, 14, 15]. Novel strategies were adopted to achieve higher levels of PC production ranging from using elicitors [16, 17], omics studies [18], mathematical modeling [19, 20], and metabolic engineering [21]. Metabolic engineering introduces a rational modification to the genetic makeup of an organism to alter or improve its metabolic profile, and consequently to develop new "non-natural" products. Engineered plant cell lines have the potential to achieve enhanced metabolite production rates [22]. Controlling these complex biosynthetic processes has been provided with the means of understanding the metabolic pathways and advances in molecular biology techniques [23]. However, the amount of PC produced by tissue and cell culture of *C. avellana* is low. A comprehensive understanding of the PC biosynthetic pathway in hazelnut, particularly the genes encoding rate-limiting enzymes and the enzymes catalyzing these reactions, are influential factors for developments in metabolic engineering and the production of PC to be significantly increased [24, 25].

The biosynthesis pathway of PC contains 19 enzymatic steps starting from the universal diterpenoid precursor called geranylgeranyl diphosphate (GGPP). GGPP is formed through coupling farnesyl diphosphate (FPP) to isopentenyl diphosphate (IPP) by GGPP synthase (GGPPS) [26]. A few PC biosynthetic genes such as *GGPPS* (Gene Bank Accession No: EF 206343) and *CgHMGR* (Gene Bank Accession No: EF553534) have been identified in hazelnut [25, 27].

Taxa-4(5), 11(12)-diene is the initial main precursor of the PC biosynthesis pathway. This precursor is catalyzed by taxadien synthase (TS) from GGPP [28]. Taxadien is hydroxylated at the C-5 position and produces taxa-4(20),11(12)-dien-5α-ol [29]. The first acetyltransferase in the PC biosynthetic pathway is taxadien-5α-ol-*O*-acetyltransferase (T5AT) which converts taxa-4(20),11(12)-dien-5α-ol to taxa-4(20), 11(12)-dien-5α-yl-acetate. Taxa-4(20), 11(12)-dien-5α-ol is acrylated by T5AT at the C-5 position in the presence of acetyl CoA [26]. This reaction is important due to the presence of an essential branch in this position. The gene encoding T5AT with some other genes is involved upstream of the biosynthetic pathway [30, 31]. High expressions of upstream biosynthesis genes such as *Taxadien-5α-ol-O-acetyltransferase* (*TDAT*) gene in some tissues or new cultivars of *Taxus* are assumed as the leading cause for higher yield of PC due to the catalysis of a series of key acetylation and hydroxylation steps [32, 33]. Furthermore, T5AT competes with taxadien-13α-hydroxylase (TαH). TαH like T5AT converts taxa-4(20), 11(12)-diene-5α-ol to taxa-4(20), 11(12) diene 5α, 13α-diol, however, its

next steps are unknown and low-expressed. Thus, this branch in the pathway could be targeted for metabolic engineering [34, 35]. These results point to TDAT's significance in the PC biosynthetic pathway, making it a promising candidate for PC engineering in *C. avellana* L. In this article, we present the novel information on the isolation and characterization of *TDAT* from the methyl jasmonate (MeJA)-induced cell suspension of hazelnut, with the goal of better understanding the PC biosynthetic pathway, which could be the subject of future researches.

## Materials and methods

### *In vitro* cell culture chemical and biological reagents

In this research, the plant materials were hazelnut (*C. avellana* L.) cotyledon-derived calli, and cell suspensions. For this purpose, hazelnut seeds were collected from Gilan Province of Iran, (37.1378˚ N, 50.2836˚ E). Friable calli and cell suspensions were obtained according to the cells immobilization method [36]. Briefly, the first step was sterilization of the seeds by sodium hypochlorite (5.25%) for 25 min, ethanol 70% (v/v) for two min, sodium hypochlorite (25 min) again, and finally sterile water (3 x 5 min). Almost 1 g from the induced white calli of seed cotyledons in MS medium supplemented with 0.2 mg l$^{-1}$ 6-benzylaminopurine (BA) and 2 mg l$^{-1}$ 2, 4-dichlorophenoxy acid (2.4-D) were suspended in 30 mL liquid media with the same composition. The cell mass obtained from several subcultures in liquid media was immobilized in solid media and produced friable and fast-growing calli. These calli were applied for subsequent experiments and the establishment of the next suspension. Culture medium salts and vitamins were produced from Merck (Kenilworth, New Jersey, USA) pharmaceutical company and other reagents including plant hormones and elicitors were produced from Sigma (St. Louis, Missouri, USA) chemical company.

### Elicitor treatment of cell suspension

Frequent subcultured (at least 4 times) and homogenized cell suspensions were used for the elicitor treatment. MeJA stock solution was prepared by dissolving it in 0.1% ethanol and water and then sterilizing it by filtering through 0.22 μm filters. The cell suspension was treated on the seventh day of its cultivation (start of stationary growth phase) with different concentrations of MeJA (0, 50, 100, 150 μM) with three replicates. In the case of control cultures (0 μM MeJA), 10 μL filter-sterilized 0.1% ethanol were added. The elicited cells were harvested after 72 h [17]. Subsequently, the harvested cells were stored in liquid nitrogen for being used in the next molecular studies.

### EST assembly strategy for identification of *TDAT*

The *TDAT* coding DNA sequence (CDS) was obtained from *C. avellana* Expressed Sequence Tags (EST) library (https://www.ncbi.nlm.nih.gov/genbank/dbest/). In brief, ESTs were downloaded from several RNA sequence projects (SRA) of hazelnut and used to build up consensus sequences. The EST sequences of *TDAT* genes were obtained from similar orthologous sequences in other plants utilizing offline BLAST software. These EST sequences were assembled through the "align-then assembles" approach using the Codon Code Aligner V.6.0.2 software (Codon Code Corporation, USA). Then, a *TDAT* consensus sequence was generated using the assembling operation and then emitting different ESTs. Accordingly, the sequence was cleaned up and a consensus sequence was created using Codon Code Aligner. This sequence was evaluated to find the open reading frame (ORF) and related protein using the ORF finder (https://www.ncbi.nlm.nih.gov/orffinder/).

**Table 1. PCR primers sequences for *TDAT* gene.**

| Position | Primer | Sequences |
|---|---|---|
| Outside ORF | Forward. 1 | 5'–GGGGGACAATCTCAAGTTCATT-3' |
| Outside ORF | Reverse. 1 | 5'–GTGGACCAAATTGAACGTACACC-3' |
| Inside ORF | Forward. 2 | 5'–CTGACAATCAGCTCAGAAGGAAAG-3' |
| Inside ORF | Reverse. 2 | 5'–CTAGACCAAGTTGTGACCAGTAG-3' |

BLASTp (https://blast.ncbi.nlm.nih.gov/Blast.cgi?PAGE=Proteins) was carried out to confirm the ORF and protein of the *TDAT* gene. Eventually, gene-specific primers were designed for PCR-amplification of the *TDAT* gene based on bioinformatics analysis. Primer premier (V.6.0) was used to design the primers based on the full ORF of the cDNA consensus sequence (Table 1). Interaction and characteristics of primers were investigated and confirmed with the oligo analyzer (https://www.idtdna.com/pages/tools/oligoanalyzer) V.3.1 program.

## Extraction of RNA and cDNA synthesis

The total RNA was extracted from MeJA treated hazelnut cells (calli) with a Total RNA isolation kit, DENAZ II ASIA, (cat No.: S-1010, Iran). The concentration and quality of the extracted RNA were analyzed using a Nanodrop spectrophotometer (OneC, Thermo Scientific, USA) and confirmed with agarose gel electrophoresis. Genomic DNA content was removed from the extracted RNA with RNase-free DNAse I (Thermo Scientific (Fermentase), cat no.: ENo521, USA). The first-strand cDNA synthesis reaction was accomplished using a Revert Aid first-strand cDNA synthesis kit (Thermo Scientific (Fermentase), cat no.: K1621, USA).

## RT-PCR analysis and gene isolation with nested-PCR

Since gene identification and isolation from cDNA depends on its expression, the cell suspension elicited with MeJA was used for RNA extraction and cDNA synthesis. Different MeJA concentrations were used to induce *TDAT* expression, and the induced cell mass with 150 μL of MeJA was used for the subsequent RT-PCR and nested-PCR. *TDAT* gene expression was investigated using reverse transcription PCR (RT-PCR). Nested-PCR was also performed to confirm the characterized ORF. Nested primers were designed in different situations of the consensus sequence. These primers were used to amplify overlapping fragments in order to approve the ORF specification (S1 Fig). The RT-PCR experiment was carried out using Forward.1 and Reverse.1 primers (Table 1). These primers amplified a fragment with a length of 1423 bp and an ORF of 1302 bp. The following reagents were used for RT-PCR and nested-PCR: 1 μL DNase-free water, 1 μL forward primer (10 pmol μL$^{-1}$), 1 μL reverse primer (10 pmol μL$^{-1}$), 2 μL cDNA (100–200 ng μL$^{-1}$), and 10 μL Tag DNA polymerase Mix Red– 1.5 μM Mgcl$_2$ (Ampliqone, cat no.: A150303, Denmark). A volume of 20 μL of each reaction was used for PCR. The RT-PCR program, as well as nested-PCR, were executed as follows: pre-denaturing cDNA at 94°C for 1 min, 34 cycles for denaturing at 94°C for 30 s, annealing at 52± 2°C for 35 s, and 85 s of extension at 72°C. After the last cycle, the amplification was extended at 72°C for 5 min. Annealing at 58°C for 35 s was set for the RT-PCR program with GAPDH primers (S1 Table).

PCR purification was done using Ron's Gel Extraction Kit (BIGRON, cat no.: 802501) for 1423 bp. The cleaned PCR products were subjected to sequencing (Bioneer, South Korea). The nested–PCR amplification and full-length cDNA sequencing were repeated four times.

## Bioinformatics analysis and modeling of the T5AT protein

The results of cDNA sequencing were analyzed using Chromas V.1.14 software. Overlapping sequences were edited using the CLC sequencing V.6.1 program, and the final sequence was used for subsequent studies. Protein sequences that identified with more than 70% of the coding region of consensus sequences from different species were selected for sequence alignment and phylogenetic analysis. The relationship between T5AT and proteins downloaded from BLASTp was determined using the web-based Clustal Omega software (http://www.ebi.ac.uk/Tools/msa/clustalo/), and the maximum likelihood was inferred using the MEGA V.7.0. A Dayhoff model was used to build the phylogenetic construction [37]. To calculate bootstrap (BS) values, 1000 iterations were used. The previously aligned nucleotide sequences of 15 genes were used to construct the evolutionary tree of the *TDAT* gene, based on the branch-site model, by using the EasyCodeML software, a part of the PAML package [38]. The branches' lengths and Omega ($\omega$) values were evaluated using Nei & Gojobori method [39] and Bayes Empirical Bayes (BEB) analysis [40].

## Secondary structure prediction of the T5AT protein

The secondary structure of the T5AT protein was predicted and assessed using the software. The conserved area of the T5AT gene was identified using the alignment of ortholog sequences for more precision and confidence. InterPro (https://www.ebi.ac.uk/interpro/) was used to represent the domains of T5AT. The PSORT-program (www.genscript.com/tools/wolf-psort) was used to determine the functional location of the T5AT protein. The position of each amino acid in the secondary structure of the protein was determined using the online POR-TER (http://distillf.ucd.ie/porter/) and PSIPRED (http://bioinf.cs.ucl.ac.uk/psipred/) programs. The protein storage site and gene ontology of T5AT were performed using PredictNLS (https://predictprotein.org/). This program showed the connection between amino acids and other proteins and their molecular functions. The Signal Peptide program (http://www.cbs.dtu.dk/services/SignalP/) was performed to predict the signal sequence of T5AT. Epestifind software (https://emboss.bioinformatics.nl/cgi-bin/emboss/epestfind) was used to rapidly identify the PEST motif (proline (P), one aspartate (D), glutamate (E), and at least one serine (S), or threonine (T)). ProtParam (https://web.expasy.org/protparam/) and ProtScale (https://web.expasy.org/protscale/) computed various physicochemical properties such as molecular weight, theoretical PI, amino acids composition, instability index, and hydrophobicity scales [41, 42].

## T5AT tertiary structure prediction

The tertiary structure of the T5AT protein was assessed using the program described herein. The amino acid sequence associated with the taxadien-5α-ol-*O*-acetyltransferase enzyme was prepared from the protein sequence database of NCBI with AAB41811 code. The T5AT sequence was compared to the sequences of proteins available in the Protein Database (PDB) to find the appropriate template creating its tertiary structure. The template crystallography file of protein (PDB ID: 4g0b) was obtained from a protein data bank. Modeler software (V.9.20) was used to generate 500 structural models of the T5AT protein. Verify3D and ERRAT compared the similarity of the simulated structure to the template. Procheck and Prosa software was used to evaluate the quality of the space chemistry models. Procheck was used to assess the overall accuracy of the protein structure and the simulated models. For this purpose, parameters such as the quality of the Ramachandran plot were examined [43]. Finally, the similarity of the 3D-structure of the template with the selected model was

estimated by measuring the root mean squared deviation (RMSD) using Swiss-Pdb Viewer (SPDBV) software (version 4.1; https://spdbv.vital-it.ch/).

## Results

### Elicitor treatment of cell suspension and RT-PCR analysis

In the primary studies, no gene (*TDAT*) expression was observed in the control (non-elicited) calli or cell suspensions (S2 Fig). MeJA elicitor treatments (50, 100, and 150 μM) were then used to induce *TDAT* gene expression. *TDAT* gene expression could not be detected at low MeJA concentrations (50 and 100 μM) and it was perceivable at higher concentration of MeJA (S2 Fig). Accordingly, 150 μM MeJA-induced cells were used for RNA extraction and RT-PCR preparation. At this concentration, PCR purification was carried out for 1423 bp and was subjected to sequencing.

### Identification and characterization of the *TDAT* gene with EST assembly and nested-PCR

First, a consensus sequence of *TDAT* was made of 152 ESTs. The ORFs and related proteins of the consensus sequence were evaluated with ORF finder and confirmed with BLAST(P). Nested-PCR was performed as a supplementary experiment to amplify the full-length cDNA of the *TDAT* from *C. avellana* and merely to further confirm the results of the bioinformatics analysis. It amplified the fragments located at the coding regions of *TDAT* with the designed primers (S3 Fig). The results of nested-PCR were precisely the same as what was expected from the bioinformatics analysis. The cDNA was 1423 bp and, according to ORF finder, it was included in 1302 bp encoding 433 amino acids (S2 Table) (submitted to GeneBank as accession number: *TDAT* (*TAT*): MF134435).

### Bioinformatics analysis and modeling of T5AT protein

To draw the phylogenetic tree and protected regions, the alignment of amino acids was carried out using Clustal Omega software. The HXXXD motif was very stable and highly conserved. Its amino acid was at positions 164–168 of *C. avellana* (Fig 1).

The results of phylogenetic analyses suggested that the *TDAT* gene in *C. avellana* is more closely related to *HHT* in *Quercus suber* and it belongs to the same order as *Quercus suber* and *Juglans regia* (Fig 2).

One of the statistical parameters to genetically evaluate the evolution process is Omega (*ω*) value, the rate ratio of nonsynonymous to synonymous substitutions ($d_N/d_S$). $\omega = 1$ suggests neutral expectation; $\omega < 1$ indicates negative (purifying) selection; while $\omega > 1$ shows positive (diversifying) selection [39]. As shown in S4 Fig, the evaluated species are divided into three distinct categories according to the evolution of the *TDAT* gene. This gene has the same ancestor in *Corylus*, *Quercus*, and *Juglans*. The assessment of nucleotide changes that alter the amino acids ($d_N$) relative to nucleotide changes that do not affect the resulting amino acid is a practical and highly efficient method to detect the process of natural selection during gene evolution. According to the matrix obtained from the Nei & Gojobori method, $d_N$, $d_S$ and *ω* were 0.0838, 0.4804, and 0.1745, respectively, for *C. avellana* L., indicating a purifying selection for the *TDAT* gene. Several statistical methods have been developed based on codon substitution models to identify positive selection for a branch in a phylogeny. One of the simulation studies in the phylogenetic analysis by maximum likelihood (PAML), using the maximum-likelihood methods is Bayes Empirical Bayes (BEB) analysis. It has been inferred from the results that Thr106 in *C. avellana*, had been influenced under positive selection by approximately 98%.

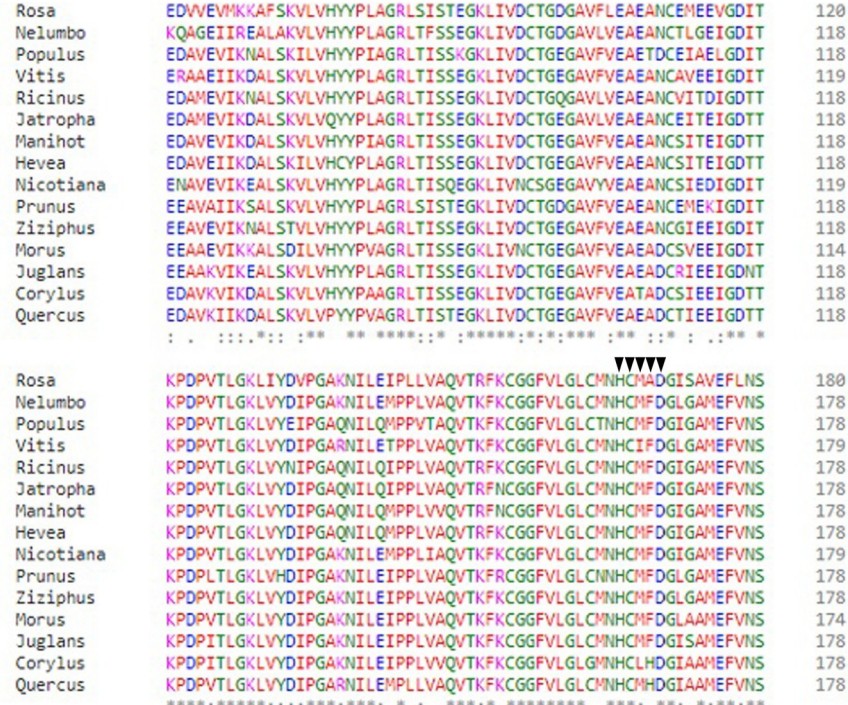

**Fig 1. Multiple sequence alignment amino acids of *TDAT* gene with other plants.** The HXXXD motif was indicated with the ▼ symbol above the motif letters.

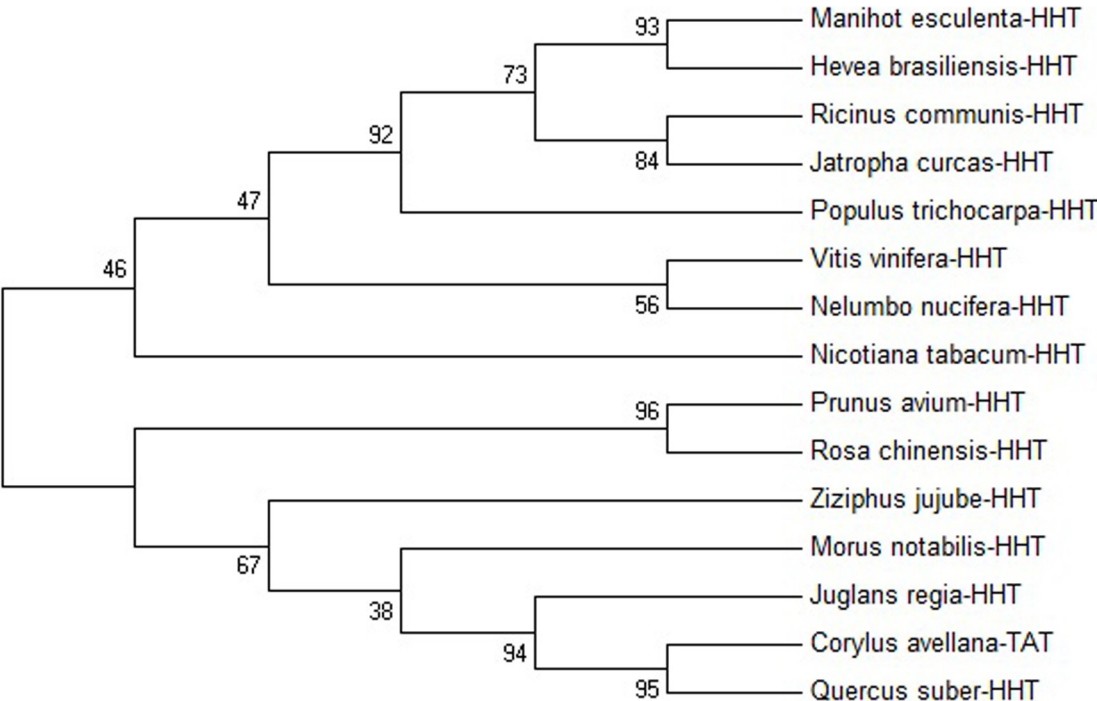

**Fig 2. Phylogenetic analysis of identified *TDAT* of *C. avellana* with *HHT* to another plant that had high identity.**

The probability of changes in amino acids Gly24, Phe32, Phe34, Pro46, Lys47, Gly50, Ala79, Gly161, Gln206, Val242, Ser307, and Pro376 were between 50% and 79%. The length of the branches showed the number of nucleotide substitutions per codon (S4 Fig). As a result, most nucleotide changes were observed for *Juglans* (0.27151) in comparison with *Corylus* (0.26287) and *Quercus* (0.22245).

## Secondary structure prediction of T5AT protein

The results of protein domain analysis, performed using the InterPro database and tools, predicted that the T5AT protein would belong to the family of transferases. The homologous superfamily of the T5AT protein has two domains. The position of the first domain was found to be at amino acids 7–216 and the second domain at 220 to 432. The molecular function of the enzyme was foreseen as the transmission of the acyl group using PredictNLS. The NLS analysis revealed the cytoplasmic localization of this protein. This software confirmed the molecular function predicted by InterPro. All protein binding sites (PBSs) which catalysis the activity and the biological process of this protein in the biosynthesis of the metabolites, were shown by PREDICT NLS (S5 Fig). The PBSs were located at amino acids 1–4, 53–56, 116–120, 212–213, 397–397, and 408–410.

Based on the Signal Peptid program, no signal peptide was determined for the T5AT protein. According to Porter's data prediction, the T5AT protein is formed through the contribution of 29.6% Helix, 26.1% extended or Beta strand, and 44.34% coils (Fig 3). The secondary structure of the T5AT protein was also confirmed with the same result provided by the PSIPRED program with a confidence line having 0 to 9 digits. The PSIPRED prediction of the location of the T5AT protein was performed in 6 positions with different scores. The active site of this protein in the plant cell was predicted to be in the cytoplasm (50%), nucleus

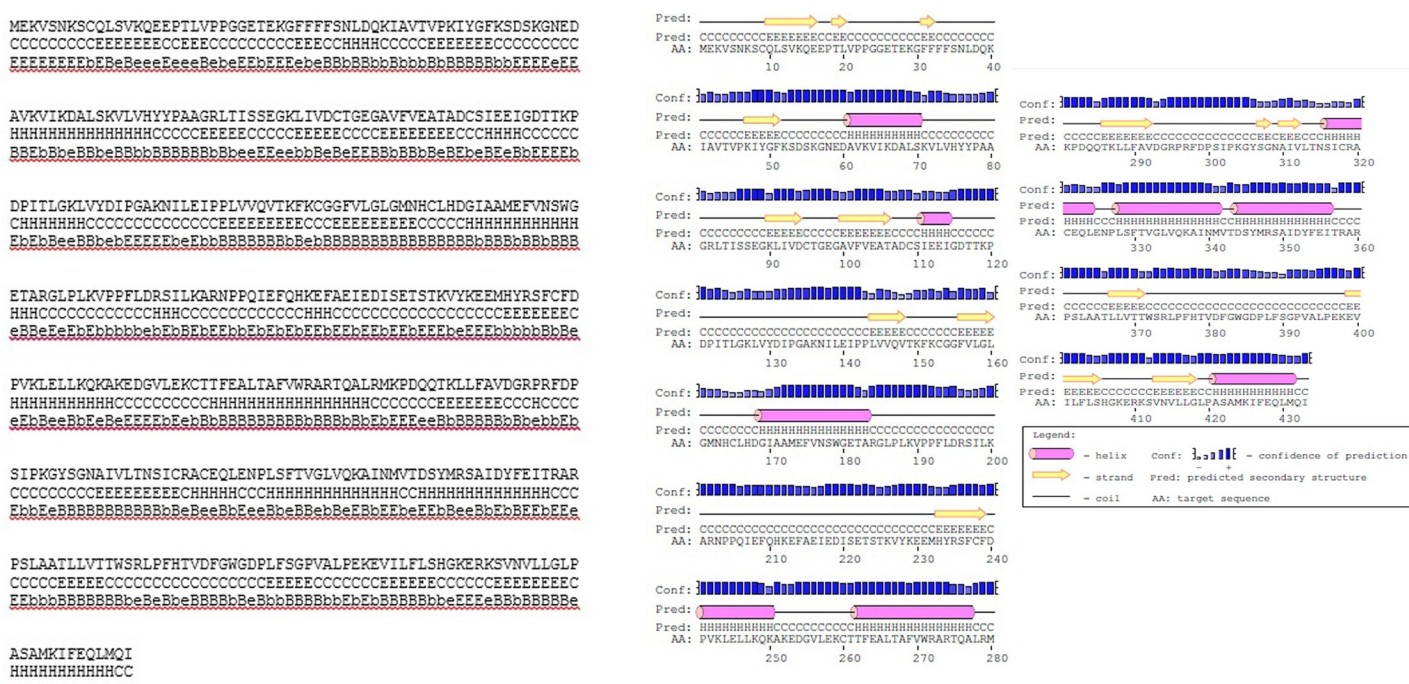

**Fig 3. The secondary structure of T5AT protein prediction with PORTER program.** Query-length: 433. H = Helix. E = strand. C = Coil. B = much buried. b = somewhat buried. e = somewhat exposed. E = very exposed.

(21.5%), mitochondrion (7.1%), peroxisome (7.1%), endoplasmic reticulum (7.1%), and chloroplast (7.1%).

Epestifind software was used to find the PEST motifs as potential proteolytic cleavage site. Altogether 4 PEST motifs were identified in the T5AT protein located between positions of 1 and 433. A potential PEST motif with 14 amino acids was determined among the positions of 14 to 29, and three poor PEST motifs were considered to be among the amino acid positions of 136–150, 319–337, and 378–398 (S6 Fig).

Based on the estimation of Protparam software, the molecular weight of the TDAT protein and its theoretical pI were 48017.62 Da and 6.43, respectively (Table 2). The result of Protscale analysis showed that T5AT had 51 positive electric charges and 53 negative electric charges, due to the presence of lysine/ arginine and aspartate/glutamate. The detailed frequency of amino acids participating in the T5AT structure is provided in Table 2. The molecular formula of this protein was assigned as $C_{2171}H_{3434}N_{560}O_{628}S_{18}$. The instability index for T5AT was estimated to be 37.94 implying the protein's stability. The Grand Average of Hydropathicity (GRAVY) value was -0.090. Furthermore, the results of the Kyte & Doolittle plot (Fig 4), as well as the tertiary structure, showed that T5AT protein fragments had 24 hydrophobic regions with amino acids 33–35, 41–47, 64–66, 68–77, 81–83, 87–89, 91–107, 124–132, 135–150, 152–163, 168–176, 189–191, 194–197, 240–243, 264–270, 288–292, 308–317, 328–339, 352–355, 363–373, 378–379, 382–393, 397–405, and 414–429. Pro 142 and Lys 56 exhibited the highest (2,033) and the lowest (-2,644) hydrophobicity residue, respectively.

**Table 2. A glance at the prediction of simple physicochemical specifications (amino acids composition) of the T5AT protein.** The amount of amino acid leucine (Leu) in the T5AT protein structure was the most.

| Amino acids | numbers | Amount (%) |
| --- | --- | --- |
| Ala (a) | 30 | 6.9% |
| Arg (R) | 16 | 3.7% |
| Asn (N) | 12 | 2.8% |
| Asp (D) | 21 | 4.8% |
| Cys (C) | 9 | 2.1% |
| Gln (Q) | 14 | 3.2% |
| Glu (E) | 32 | 7.4% |
| Gly (G) | 29 | 6.7% |
| His (H) | 7 | 1.6% |
| Iis (I) | 25 | 5.8% |
| Leu (L) | 42 | 9.7% |
| Lys (K) | 35 | 8.1% |
| Met (M) | 9 | 2.1% |
| Phe (F) | 25 | 5.8% |
| Pro (P) | 27 | 6.2% |
| Ser (S) | 27 | 6.2% |
| Thr (T) | 26 | 6.0% |
| Trp (w) | 4 | 0.9% |
| Tyr (Y) | 9 | 2.1% |
| Val (V) | 34 | 7.9% |
| Pyl (O) | 0 | 0.0% |
| Sec (U) | 0 | 0.0% |

Number of amino acids: 433.

Molecular weight: 48017.62.

Theoretical pI: 6.43.

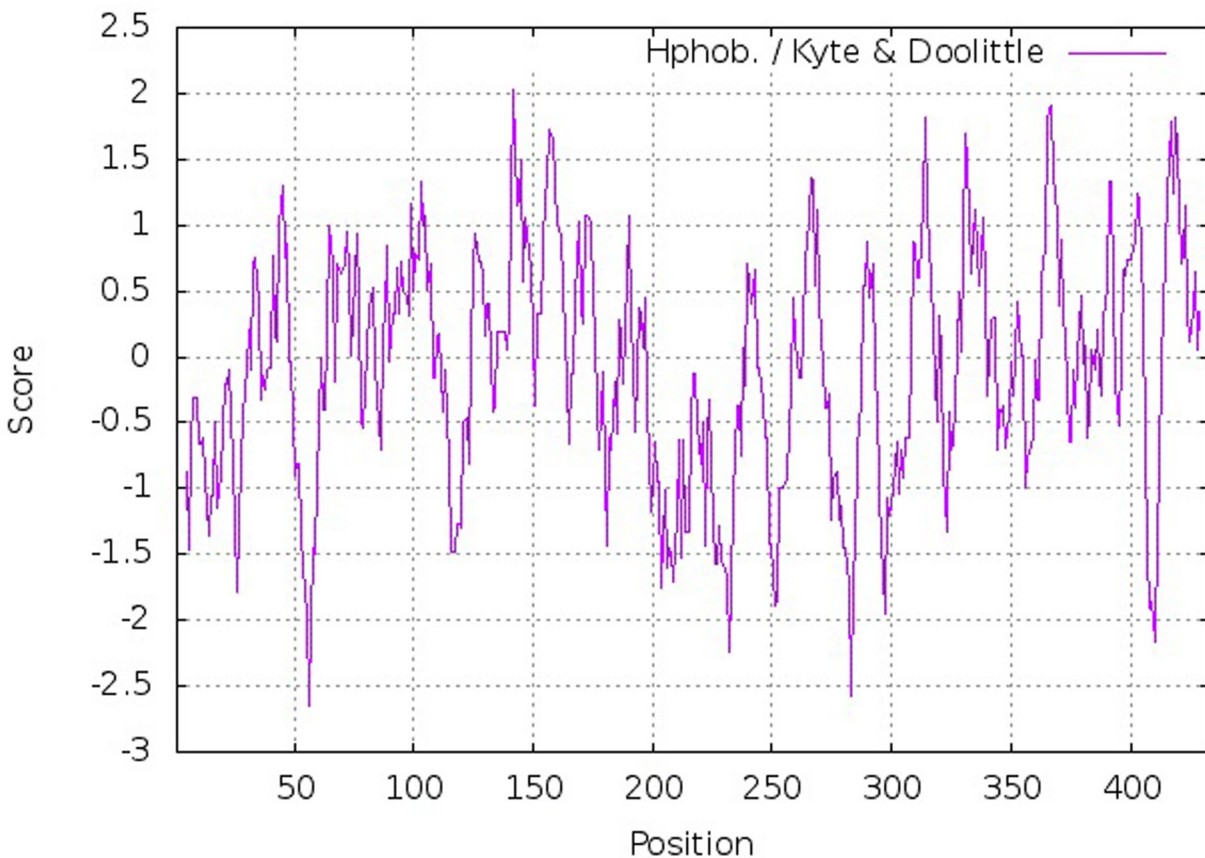

**Fig 4. The results of Kyte and Doolittle diagram about the T5AT protein containing 24 hydrophobic regions.**

### 3D-structure of the T5AT protein

First, similar and most close sequences to the T5AT sequence were obtained using the protein BLAST database. The hydroxycinnamoyl-COA shikimate/quinate hydroxycinnamoyl transferase enzyme, which it's 3D-structure (PDB code; 4G0B) was identified by Lallemand *et al*. [44], was considered as a template for the T5AT protein. The sequence identity between the two proteins was determined to be 35%. Five hundred structural models were generated T5AT using Modeller software. Validation tests evaluated the quality of each model, and the model with the highest degree of similarity was considered as the tertiary structure of the T5AT protein which was referred to as MO323. The results of Ramachandran plots (S7 Fig) were summarized in Table 3. Approximately, 87.0% and 88.0% of all amino acids were located within the most favored regions for the template and MO323, respectively. The slight difference between the obtained values can be explained by the position of amino acids in specific regions of the Ramachandran plot, implying that the 3D-structure of MO323 was highly similar to that

**Table 3. Related parameters for confirmation of the simulated structure and its similarity to the pattern.**

| Model | Procheck | | | | Verify 3D | Errat |
|-------|----------|---|---|---|-----------|-------|
| | **Most favored regions** | **Additional allowed regions** | **Generously allowed regions** | **Disallowed regions** | | |
| **4G0B** | 87.0% | 12.7% | 0.3% | 0.0% | 89.46% | 90.3061 |
| **MO323** | 88.8% | 10.4% | 0.8% | 0.0% | 60.51% | 75.5882 |

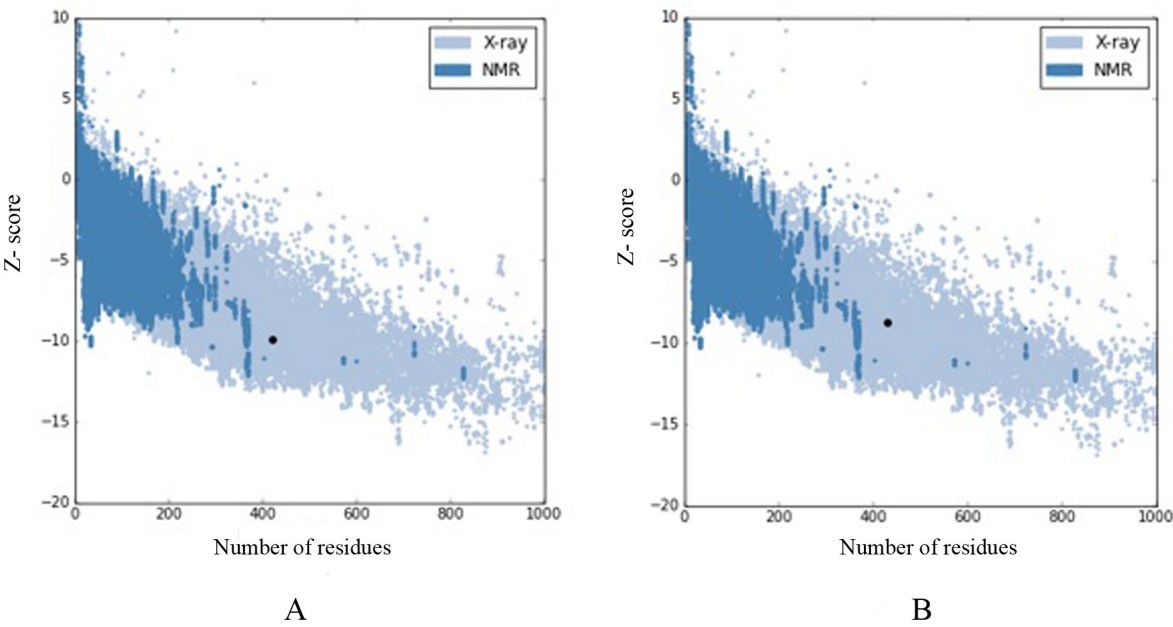

**Fig 5.** Z-score values for (A) the template, (B) the simulated model (MO323). Both structural models are located in the position of X-ray structures.

of the template. The overall quality of the MO323 was checked by comparing Z-scores in Prosa software. The Z-score calculates the total energy of the structure and shows the degree of consistency between the sequence and the tertiary structure of the model [45]. This value was calculated as -9.94 and -8.72 for the template and MO323, respectively. The negative values represented the accuracy of the simulated structure; accordingly, there was a high similarity between MO323 and the template. As shown in Fig 5, the structure of MO323 was located at the position corresponding to the X-ray structures. Thus, the obtained model was sufficiently reliable and was empirically close to the 3D-structure of the pattern. Verify3D calculations showed that 60.51% of the amino acids belonging to MO323 had a score greater than 0.2. This value was 89.46% for the template. This parameter determines the compatibility of the protein 3D-structure with its 1D-structure. Based on the obtained score, the accuracy of the simulated structure and its high quality can be inferred. ERRAT values were determined to be 75.8 and 90.5 for MO323 and the template, respectively. This parameter examines non-bonded interactions between different atom types and the performance rates of 9 amino acids versus other 9 amino acids by statistically comparing with the most similar structures. Based on the results, MO323 was selected as the T5AT 3D-structure from 500 models. The RMSD values for the template and the model (based on backbone and α-carbon) were 0.79 and 0.81 angstroms, respectively. The slight differences between the RMSD values of the two proteins demonstrated their structural similarity. The 3D-structures of the template and MO323 were merged, and the overall similarity and slight differences between them were examined using YASARA software. The results showed that the topology of both proteins' folding was significantly similar despite the slight differences between the model and the template (Fig 6). Differences were more specified in the turn and coil regions. As shown in Fig 7A, the T5AT enzyme contained 15 beta-strands and 9 alpha-helices in its tertiary structure. Beta-strands were located at the amino acid positions 12–21, 45–51, 30–32, 82–86, 89–95, 100–107, 143–151, 154–163, 231–239, 281–294, 310–318, 368–372, 388–392, 399–405, and 412–420. Alpha-helices were located at amino acid positions with 35–40, 61–72, 168–184, 240–249, 261–278, 319–326, 328–341,

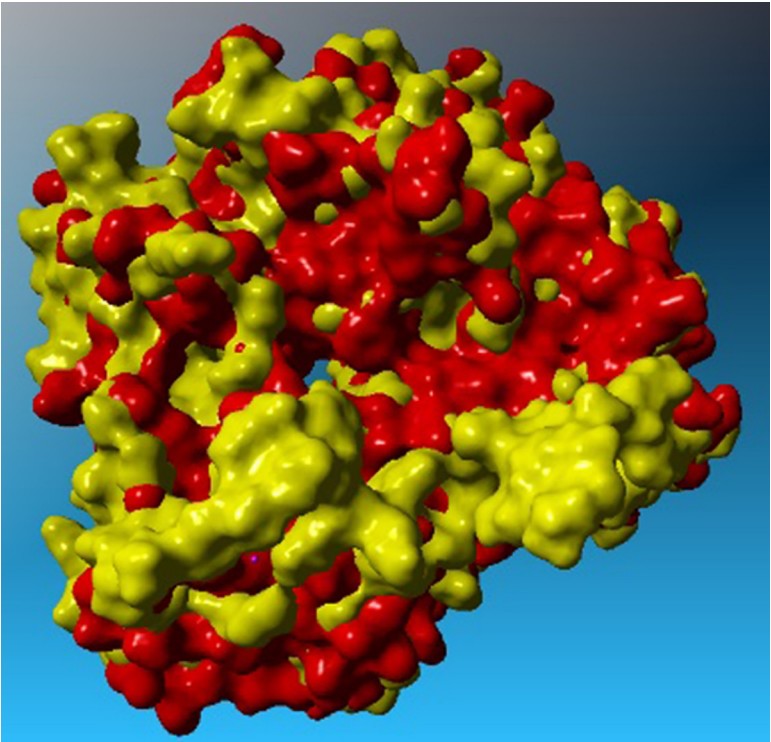

**Fig 6. Comparison of the surface of the 3D-structure belongs to the template and MO323 (red and yellow, respectively), differences of structures are relative to turn and coil regions.**

343–355, and 423–432. The T5AT enzyme belongs to the transferase superfamily which is commonly found in plants and fungi. The HXXXD motif makes up part of the active site of all enzymes in this superfamily [46]. This motif located at positions 164–168 of the T5AT protein structure contains the His-Cys-Leu-His-Asp amino acid sequence forming a turn region between the eighth beta-strand and the tertiary alpha-helix (Fig 7B).

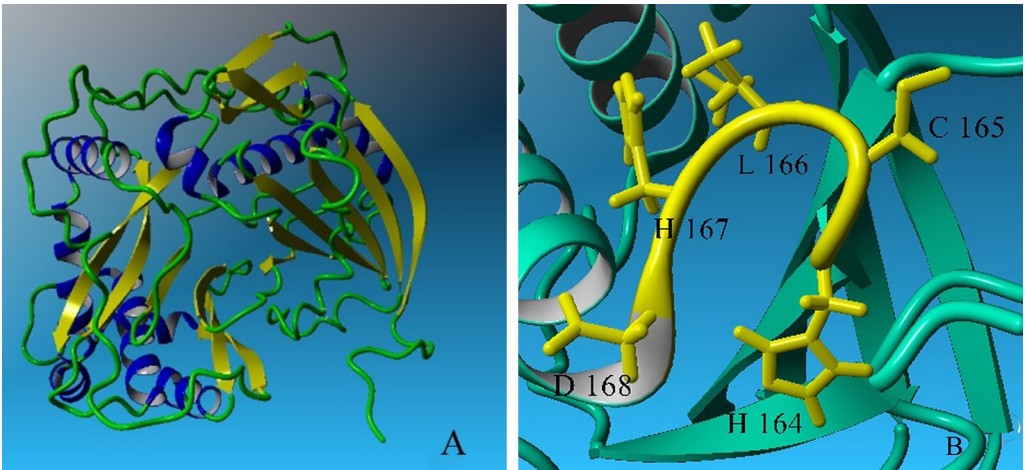

**Fig 7. 3D-structure of the T5AT enzyme presented by YASARA program.** (A) Beta-strands and alpha-helixes are yellow and blue, respectively. This model has 15 beta-strands and 9 alpha-helices; (B) HXXXD motif in T5AT enzyme contains His 164, Cys 165, Leu 166, His 167 and Asp168.

## Discussion

A considerable number of research papers have been devoted to investigating the taxol biosynthetic pathway in *Taxus* and its producing microorganisms [29, 47]. Despite all these efforts, that pathway still has ambiguities in details and it is yet unknown in hazelnut except for the early two genes [25, 27]. The key acetylation step by *taxadien-5α-ol-O-acetyltransferase* (*TDAT* or *TAT*), as the early bottleneck in the pathway, has been highlighted [48]. Moreover, the key genes participating in the next hydroxylation steps, are affected by the *TDAT* catalytic step [32]. In this study, *TDAT* ortholog was detected and identified in *C. avellana* L. using the experimental and *in-silico* analysis. *C. avellana* is known as a candidate source for PC production [49]. The full-length cDNA encoding *TDAT* from *C. avellana* (gene accession number: MF134435) was 1423 bp, similar in size to *omega-hydroxypalmitate o-feruloyl transferase* (*HHT*). The cDNA of *TDAT* contains a 1302 bp ORF encoding a protein of 433 amino acids. The constructed phylogenetic tree indicated a high degree of identity between *TDAT* and *HHT*. *Corylus*, *Quercus suber*, *Juglans regia*, *Morus notabilis*, and *Ziziphus jujube* resided on the same branch of the phylogenetic tree. It should be noted that each of these plants belongs to different families, indicating this gene was protected in distant families. Early clarifying assessments revealed that *TDAT* is also multi-specific in *Taxus* [48, 50]. Investigation on the *TADT* genetic evolution showed that Thr106 had a positive change of 98% compared to other species. The coding sequence of this amino acid for *Corylus* is ACA, while other species have GAA (12 cases) and GAG (2 cases) for Glu at this position. Thr106 was positioned in the beta structure of the T5AT 3D-structure and away from the enzyme active site. Therefor altering this amino acid probably has little effect on the active site. Analysis of the enzyme 3D-structure showed that Gly24, Phe32, Phe34, Pro46, Lys47, Ala79, Gly161, Ser307, and Pro376 are close to the active site, and altering of these amino acids might affect their function.

The *taxadien-5α-ol-O-acetyltransferase* gene has been identified in the yew, another main source of PC. This gene, called *TmTAT*, was known in *T. media*. The ORF of the *TmTAT* gene was 1317 bp encoding 439 amino acids [51]. The molecular weight of the T5AT protein was 48.17 kDa. The theoretical pI of T5AT was 6.43, and the instability index for the protein was 37.94. The result of BLASTp indicated that the T5AT protein was closely similar to the HHT protein in *Juglans regia* and *Ziziphus jujube*. The isoelectric point of HHT in *J. regia* and *Z. jujube* was 5.83 and 5.66, respectively. The instability index of HHT in *J. regia* and *Z. jujube* was calculated as 39.31 and 35.75, respectively. The molecular weight of the HHT protein was very close to that of the T5AT protein—approximately 48 kDa. The HHT protein consisted of 432 amino acids in both plants.

To investigate the evolutionary relations between the transferase in *Taxus* and *Corylus*, a phylogenetic tree was constructed based on the amino acid sequences of TmTAT and transferase from other *Taxus* species. The results highlighted that TmTATs, (T5ATs) from *T. chinesis* and *T. cuspidata* were grouped into the cluster with the shortest distance from the T5AT of *C. avellana* while the TmTAT from *T. canadensis* was categorized into a cluster farther away from *C. avellana*. Moreover, the T5ATs from *C. avellana* and *Taxus* species were found to be dissimilar [24]. The *taxadien-5a-ol-O-acetyl transferase* ORF in *T. chinensis* had 1275 nucleotides that encodes 425 amino acids. The protein had an isoelectric point of 5.39 and a molecular weight of 47 kDa [52]. The molecular weight of the TmTAT was 49 kDa, and its instability index was 35, similar to the other *Taxus* species. On close scrutiny, *TmTAT* was strongly expressed in the leaves, weakly expressed in the stems, and had no expression in the fruits of *T. media*. A detailed look at this tissue expression pattern revealed that this is a tissue-specific gene [24]. The T5AT protein had two domains as did the HHT protein in *Quercus suber*. While, the position of the first domains of both proteins were located at amino acid positions

7–216, but the second domains were found at different positions. In contrast, the second domain of HHT was located at amino acid positions 221–431, which differed in the position of just one amino acid in T5AT (220 to 430). However, these second domains indicated that T5AT and HHT belong to the superfamily of transferase enzymes [24, 52].

The T5AT and HHT proteins, as well as most acetyltransferase enzymes, have a functional HXXXD motif. This motif resided in amino acid position 164 [46]. In *T. chinensis*, this protein has a functional HXXXD motif with the function of transferring acetyl from the precursor of the CoA-acetyl and additionally, two other domains at positions 7–214 and 223–431 [52]. Based on the NCBI database, the TmTAT protein has a second domain belonging to the family of the transferase, a functional HXXXD motif, which plays a role in the transmission of the acetyl group, and also two domains located at amino acid positions 8–214 and 223–431 [24]. It was revealed that the HXXXD motif is probably part of the active site [46]. The previous studies reported a lack of signal peptide in T5AT, HHT, and TmTAT proteins; therefore, these proteins probably act in cytoplasm. The biological process of the T5AT protein was predicated in the biosynthesis of metabolites, transferase activity, and the transmission of acetyl group. The molecular function of taxadien-5a-ol-*O*-acetyltransferase (T5AT) was identified as transmitting acetyl group from acetyl-CoA and producing the T5AT enzyme in *T. cuspidate* [34]. Furthermore, the molecular function and biological process of the TmTAT enzyme have been reported to be the same as those of T5AT in *T. cuspidate* [24]. In this study, 29.6% Helix, 26.1% Extended, and 44.34% Coils were observed in the secondary structure of T5AT with 433 amino acids, while 37% Helix, 26% Beta strand or Extended, and 37% Coils contributed to the formation of the TmTAT protein [52]. The pattern of the T5AT protein in drawing the three-dimensional structure was HOST in *Coffea canephora* (PDB Id: 4g0b). It was confirmed that His-153 in HCT is an active site, as the His-153-ALa mutation had no enzymatic activity as a catalytic enzyme. The His-153 in HCT from the conserved HXXXD motif was confirmed as a catalytic residue by mutagenesis [53]. Therefore, His-164 is found in the active site of the T5AT protein (*C. avellana*). The active sites in the T5AT in *T. cuspidate* were also included His-164 and Asp-373. As a result, it is most likely that Asp-381 is another amino acid in the active site involved in the T5AT (*C. avellana*). ALa-156-Ser mutants produced 4-fold more diCQAs. Ser is a polar amino acid with an OH group. Since His is a polar amino acid, it is expected that His-167 will improve enzyme activity [44]. The HST protein pattern in *Arabidopsis thaliana* (PDB Id: 5kjs) was used to draw the tertiary structure of the HCT protein in *C. canephora*. HCT and HST proteins are located in the BAHD acyltransferase family. This protein family is recognized by the sequence homology and HXXXD and DFGWG conserved motifs and considered as the main enzymes in a wide range of secondary metabolites' biosynthesis pathway with a wide variety of substrates. In the active site, the HHAAD and DFGWG contain the two essential amino acids His153 and Asp380. [26, 44]. Our findings show that these two motifs are found in the HCT structure in the regions 157–157 and 381–385, with the same sequence. The first motif, HCLHD, is found at position 164–168 in the T5TAT structure sequence, and the second motif, which has a completely identical amino acid sequence to HCT and HST, is found at position 381–385 in the protein. The existence of the conserved motifs in the structure of T5TAT protein indicates that it belongs to the BAHD acyltransferase family and transferase superfamily.

## Conclusion

Bioinformatics research was used to identify and characterize taxadien-5-ol-O-acetyltransferase (TDAT), which catalyzed the third stage of PC biosynthesis in *C. avellana*. Our success in finding this gene was due to the use of elicitor to induce gene expression, as well as the use of

the EST library and bioinformatics studies. It can be assumed that the taxadien-5α-ol-*O*-acetyltransferase enzyme has undergone numerous changes during its evolution. Although this enzyme varies significantly from *Taxus'* T5AT enzyme, their active site, and conserved motif are identical. It also has similar performances in different plants. This gene locates in the early steps of the PC biosynthetic pathway. On their previous substrate, taxadien-5-ol, T5AT competed with T13Ha. As a result of its location at the start of the pathway and on the branch, this gene is crucial. This is a very low-expression gene that necessitates the use of high-level elicitors for its identification. Herein, for the first time, the full length and features of the *TDAT* were identified using bioinformatics analysis rather than costly, time–consuming, and labor-intensive work. Understanding the sequence and characteristics of *TDAT* will offer promisingly to PC engineering programs and maximize its production in hazelnut for future studies.

## Supporting information

**S1 Fig. The position of full-length cDNA and nested-PCR primers containing *TDAT* gene.** The arrows define the primers and the green rectangles illustrate nested-PCR products for each primer.
(DOCX)

**S2 Fig. Agarose (1%) gel electrophoresis.** Evaluation of RT-PCR product of the different concentrations of MeJA induced cells and PCR purification. L: (ladder 100 bp), 1: Negative control (water), 2: RT-PCR products without any treatment (control), 3: RT-PCR product with 50 μM of MeJA, 4: RT-PCR product with 100 μM of MeJA, 5: RT-PCR product with 150 μM MeJA, 6: Negative control, 7: PCR product after PCR-purification in 150 μM of MeJA. The grouping of gels which have been cropped from different gels was identified with vertical white lines.
(DOCX)

**S3 Fig. Agarose (1%) gel electrophoresis of nested-PCR analysis of *C. avellana*; Ladder (L) 100bp,1: Forward 1 + Reverse 1 (Fr$_1$+Rr$_1$), 2.** Forward 1+ Reverse 2 (Fr$_1$+Rr$_2$), 3. Forward 2 + Reverse 1 (Fr$_2$+Rr$_1$), 4. Forward 2 + Reverse 2 (Fr$_2$+Rr$_2$), 5. Positive control (GAPDH primers), 6. Negative control (water).
(DOCX)

**S4 Fig. Evolutionary relationships of *C. avellana* L. *TDAT* gene with 15 geneses, based on the branch-site model, by using the EasyCodeML software (PAML package).**
(DOCX)

**S5 Fig. The protein binding sites (PBS) showed by PREDICT NLS. All indicated parts with red diamonds are PBS.** The regions are 1–4, 53–56, 116–120, 212–213, 397–397, and 408–410.
(DOCX)

**S6 Fig. The position of the PEST motif was detected with Epestifind software.**
(DOCX)

**S7 Fig. The Ramachandran plot and the frequency of amino acids in a different position for *C. avellana* L.** (A) 4G0B template (B) MO323. In the picture, [A, B, L] show a very favored region for amino acids. [a, b, l, p] show the allowed zone, and [~a, ~b, ~l, ~p] show the permitted regions with ignorance.
(DOCX)

**S1 Table. PCR primers sequences for *GAPCH* gene.** (Thermo Scientific (Fermentase), cat no.: K1621, USA).
(DOCX)

**S2 Table. The results of ORFs, online software.** The most portable framework and the tallest of ORF are in + 1 and the nucleotide of 97 to 1398.
(DOCX)

**S1 Raw images.**
(PDF)

## Author Contributions

**Data curation:** Sajad Rashidi Monfared.

**Formal analysis:** Mona Raeispour Shirazi, Sara Alsadat Rahpeyma, Sajad Rashidi Monfared, Jafar Zolala, Azadeh Lohrasbi-Nejad.

**Investigation:** Mona Raeispour Shirazi.

**Methodology:** Sara Alsadat Rahpeyma, Sajad Rashidi Monfared, Jafar Zolala, Azadeh Lohrasbi-Nejad.

**Project administration:** Mona Raeispour Shirazi.

**Resources:** Jafar Zolala.

**Software:** Mona Raeispour Shirazi, Sajad Rashidi Monfared, Azadeh Lohrasbi-Nejad.

**Supervision:** Sara Alsadat Rahpeyma.

**Validation:** Sajad Rashidi Monfared, Jafar Zolala.

**Writing – original draft:** Mona Raeispour Shirazi.

**Writing – review & editing:** Sara Alsadat Rahpeyma, Sajad Rashidi Monfared, Jafar Zolala, Azadeh Lohrasbi-Nejad.

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
