## [Decision Letter · Decision Letter 0]

22 Apr 2021

PONE-D-21-05459

Identification and in-silico characterization of taxadien-5α-ol-O-acetyltransferase (TDAT) gene in Corylus avellana L.

PLOS ONE

Dear Dr. Rahpeyma,

Thank you for submitting your manuscript to PLOS ONE. After careful consideration, we feel that it has merit but does not fully meet PLOS ONE’s publication criteria as it currently stands. Therefore, we invite you to submit a revised version of the manuscript that addresses the points raised during the review process.

We look forward to receiving your revised manuscript.

Kind regards,

Balamurugan Srinivasan

Academic Editor

PLOS ONE

Journal Requirements:

Reviewers' comments:

Reviewer's Responses to Questions

**Comments to the Author**

1. Is the manuscript technically sound, and do the data support the conclusions?

Reviewer #1: Yes

Reviewer #2: Yes

2. Has the statistical analysis been performed appropriately and rigorously? 

Reviewer #1: Yes

Reviewer #2: Yes

3. Have the authors made all data underlying the findings in their manuscript fully available?

Reviewer #1: Yes

Reviewer #2: Yes

4. Is the manuscript presented in an intelligible fashion and written in standard English?

Reviewer #1: Yes

Reviewer #2: No

5. Review Comments to the Author

Reviewer #1: This manuscript presents interesting bioinformatic findings about TAT in hazelnuts species, which will be interesting to the community, specially the prediction of secondary and tertiary structures.

Nevertheless it needs some serious improvements in the literature review as well as how the data and figures are explained and presented. Some figures are blurry and couldn’t be assessed properly. It needs to compare more with recents studies where the TAT has been utilised and highlight more the relevance of the work too.

Please do find the specific comments on the pdf file attached

Reviewer #2: Abstract

Line no. 16 – 18: ‘and tissue cultures of Taxus 16 species and’ not correlating here. Reframe the sentence.

Line no. 20: change as ‘this study was undertaken to …..’

Introduction

Line no. 44 – 46: Sentence starting as ‘In particular……’ rewrite the sentence

Line no. 55 – 58: Sentence starting as ‘The deep and detailed understanding’ rewrite the sentence.

Line no. 65: Change as ‘The initial main precursor of PC’

Materials and Methods

Line no. 88: ‘were procured from Merck’

Line no. 89: were procured from Sigma’

Line no 94: dissolving

Results

Line no. 195: (non-elicited)

Line no. 196 and 197: Remove (0)

Line no. 200: change as ‘for 1423 bp and were subjected to sequencing.

Line mo. 305 – 308: consider rewriting the sentence

General Comments

The authors have to improve the language in the entire manuscript to meet the journal standard.

6. PLOS authors have the option to publish the peer review history of their article (what does this mean?). If published, this will include your full peer review and any attached files.

Reviewer #1: No

Reviewer #2: **Yes: **Muthukrishnan Arun

---

## [Author Response · Author response to Decision Letter 0]

2 Jun 2021

Dear Editor-in-Chief,

We would like to deeply thank the editorial office and the reviewers for the constructive comments which helped us to improve the paper and thank you for giving us the opportunity to submit a revised draft of our manuscript titled "Identification and in-silico Characterization of taxadien-5α-ol-O-acetyltransferase (TDAT) Gene in Corylus avellana L". The authors tried to address all reviewers’ comments in the revised manuscript. We hope our revision has improved the paper to a level of their satisfaction. Number-wise Responses to the specific editorial comments, suggestions, and queries are given below; the revisions in the manuscript have been marked highlighted with track changes in a file labeled ‘ Revised manuscript with track changes.

Enclosed, please find the revised version of the manuscript entitled: 

“Identification and in-silico Characterization of taxadien-5α-ol-O-acetyltransferase (TDAT) Gene in Corylus avellana L.

By:

Mona Raeispour Shirazi, Sara Alsadat Rahpeyma, Sajad Rashidi Monfared, Jafar Zolala and Azadeh Lohrasbi-Nejad1

Sincerely yours,

Dr. Sara Alsadat Rahpeyma

Assistant Professor of Plant Breeding

Department of Agricultural Biotechnology

Faculty of Agriculture

Shahid Bahonar University of Kerman

Iran 

Responses to journal requirements:

- The authors tried to improve and adjust the manuscript style to meet PLOS ONE’s style found at https://journals.plos.org/plosone/s/file?id=wjVg/PLOSOne_formatting_sample_main_body.pdf and https://journals.plos.org/plosone/s/file?id=ba62/PLOSOne_formatting_sample_title_authors_affiliations.pdf. Track changes were used to mark the changes, including text format, figures, tables, references, grammatical adjustments, etc, across the updated manuscript. 

- 

- Two gel figures are included in the manuscript "PONE-D-21-05459" (Fig 2. and S1 Fig). The authors attempted to compile original gel figs in a single PDF file format called "S1-row pictures," which contains all the original and unadjusted gel images, included in the manuscript’s main figures and supplemental figures. Fig 2. is in the public data presented in the main article and S1 Fig is in the ‘supporting information’ section. Also, the author tried to consider all requirements for gel reporting, such as labeling and annotating the experimental and not included lanes, molecular markers’ weight, and so on. Unfortunately, we lost all of the original gel images due to a serious problem with my student (Mona Raeispour) laptop, and we only have cropped images retrieved from emails. But I, as the corresponding author, ensure the editorial board that no manipulating has been done in figures. The cropped lanes of images were identified with the blanks among the parts. The photos were captured using the VIBER Company's Quantum gel documentation imaging technology. The authors hope that the revision will meet PLOS ONE's approval.

Response to Comments by Reviewer 1

We'd like to express our gratitude to the Reviewer for his or her constructive comments and suggestions. Based on his/her remarks, various revisions were made to improve the quality of the manuscript. The modifications were highlighted with track changes through the revised manuscript. The next sections contain our responses to the dear reviewer comments 

Comment No.1: This manuscript presents interesting bioinformatic findings about TAT in hazelnuts species, which will be interesting to the community, specially the prediction of secondary and tertiary structures. Nevertheless it needs some serious improvements in the literature review as well as how the data and figures are explained and presented.

Response: Thanks to the reviewer for the kind comment on the manuscript. The author tried to improve the literature review by including more recent studies and providing additional details about the presented study. The modifications have been addressed in the text as well as in the separate responses to the attached comments.

Comment No.2: Some figures are blurry and couldn’t be assessed properly. 

Response: We considered the reviewer’s suggestion and Figs 1, 2, 3, 6, and 9 changed to more appropriate ones. All figures were checked to meet journal style.

Comment No.3: It needs to compare more with recents studies where the TAT has been utilised and highlight more the relevance of the work too. 

Response: To highlight the significance of the subject, the recent and relevant TAT studies were added to the text throughout the introduction and discussion sections. They are addressed via track changes.

The following new references were added to the introduction and discussion:

- Salehi M, Moieni A, Safaie N, Farhadi S. Whole fungal elicitors boost paclitaxel biosynthesis induction in Corylus avellana cell culture. Plos one. 2020;15(7):e0236191.

- Yu C, Zhang C, Xu X, Huang J, Chen Y, Luo X, et al. Omic analysis of the endangered Taxaceae species Pseudotaxus chienii revealed the differences in taxol biosynthesis pathway between Pseudotaxus and Taxus yunnanensis trees. BMC plant biology. 2021;21(1):1-13.

- Salehi M, Farhadi S, Moieni A, Safaie N, Hesami M. A hybrid model based on general regression neural network and fruit fly optimization algorithm for forecasting and optimizing paclitaxel biosynthesis in Corylus avellana cell culture. Plant Methods. 2021;17(1):1-13.

- Farhadi S, Salehi M, Moieni A, Safaie N, Sabet MS. Modeling of paclitaxel biosynthesis elicitation in Corylus avellana cell culture using adaptive neuro-fuzzy inference system-genetic algorithm (ANFIS-GA) and multiple regression methods. PloS one. 2020;15(8):e0237478.

- Sanchez-Muñoz R, Bonfill M, Cusidó RM, Palazón J, Moyano E. Advances in the regulation of in vitro paclitaxel production: Methylation of a Y-Patch promoter region alters BAPT gene expression in Taxus cell cultures. Plant and Cell Physiology. 2018;59(11):2255-67.

- Sanchez-Muñoz R, Perez-Mata E, Almagro L, Cusido RM, Bonfill M, Palazon J, et al. A novel hydroxylation step in the taxane biosynthetic pathway: a new approach to paclitaxel production by synthetic biology. Frontiers in Bioengineering and Biotechnology. 2020;8:410.

- Kuang X, Sun S, Wei J, Li Y, Sun C. Iso-Seq analysis of the Taxus cuspidata transcriptome reveals the complexity of Taxol biosynthesis. BMC plant biology. 2019;19(1):1-16.

- Kanda Y, Nakamura H, Umemiya S, Puthukanoori RK, Murthy Appala VR, Gaddamanugu GK, et al. Two-Phase Synthesis of Taxol. Journal of the American Chemical Society. 2020;142(23):10526-33. doi: 10.1021/jacs.0c03592.

- Wang T, Chen Y, Zhuang W, Zhang F, Shu X, Wang Z, et al. Transcriptome sequencing reveals regulatory mechanisms of taxol synthesis in Taxus wallichiana var. Mairei. International journal of genomics. 2019.

- He C-T, Li Z-L, Zhou Q, Shen C, Huang Y-Y, Mubeen S, et al. Transcriptome profiling reveals specific patterns of paclitaxel synthesis in a new Taxus yunnanensis cultivar. Plant Physiology and Biochemistry. 2018;122:10-8.

- Zhou X, Zhu H, Liu L, Lin J, Tang K. A review: recent advances and future prospects of taxol-producing endophytic fungi. Applied Microbiology and Biotechnology. 2010;86(6):1707-17. doi: 10.1007/s00253-010-2546-y.

- Walls LE, Malcı K, Nowrouzi B, Li RA, d'Espaux L, Wong J, et al. Optimizing the biosynthesis of oxygenated and acetylated Taxol precursors in Saccharomyces cerevisiae using advanced bioprocessing strategies. Biotechnology and Bioengineering. 2021;118(1):279-93.

- Walker K, Ketchum RE, Hezari M, Gatfield D, Goleniowski M, Barthol A, et al. Partial Purification and Characterization of Acetyl Coenzyme A: Taxa-4 (20), 11 (12)-dien-5α-olO-Acetyl Transferase That Catalyzes the First Acylation Step of Taxol Biosynthesis. Archives of biochemistry and biophysics. 1999;364(2):273-9.

The specific comments on the pdf file attached 

Abstract

Comment No.1: please clarify that T5AT is encoded by TDAT gene.

Response: Thank you for making the suggestion. The phrase “, encoded by TDAT,” was added to the text. 

Comment No.2: The abstract should be improved with a nice conclusion.

Response: The author tried to create a more appropriate conclusion for the abstract.

Introduction:

Comment No.3: In general introduction is good but the novelty and the aim of study are poorly written. Therefore, introduction should be improved with a better conclusion. 

Also, the current studies on TAT characterization or production could be mentioned here.

Response: Thanks to the reviewer for the comment. The introduction now includes recent studies on TDAT and taxol that are relevant to the issue. A better conclusion has also been added to the introduction. For this aim the sentences “These results point to TDAT's significance in the PC biosynthetic pathway, making it a promising candidate for PC engineering in C. avellana L. In this article, we present novel information on the isolation and characterization of TDAT from methyl jasmonate (MeJA)-induced hazelnut cell suspension, with the goal of better understanding the PC biosynthetic pathway, which could be the subject of future research.” was added to the introduction.

Materials and methods:

Comment No.4: should not be "bought from"?

Response: The sentence “Culture medium salts and vitamins were provided from Merck” was replaced with “Culture medium salts and vitamins were produced from Merck”

Comment No.5: the same

Response: The modification is like the previous response.

Comment No.6: the link does not work, please provide the correct one

Response: The link was replaced with https://www.idtdna.com/pages/tools/oligoanalyzer. The other links within the text were checked and updated.

Comment No.7: this section needs corrections and should be rewritten

Response: The author tried to improve the section and rewrite some sentences to get more clarified.

Comment No.8: Please clarify here. Was qPCR used for gene expression detection? RT-PCR stands for reverse transcription PCR. 

Response: In this study, qRT-PCR was not used. This part was clarified with the sentence “TDAT gene expression was investigated using reverse transcription PCR (RT-PCR).

Comment No.9: should be reworded

Response: The sentence was changed to “These primers were used to amplify overlap fragments in order to approve the ORF specification” 

Comment No.10: The primer names should be same as the table

Response: The name of primers replaced with “forward. 1 and reverse. 1” the same as other places within the text. 

Comment No.11: Fig S1 should be mentioned here

Response: The modifications have been considered.

Comment No.12: The figure caption should be reworded

Response: The modifications have been considered and “the situation of cDNA full length and nested-PCR primers comprising TDAT gene.” was replaced with “The position of full-length cDNA and nested-PCR primers containing TDAT gene”. 

Comment No.13: The links given should be corrected, none of them is working. Some of them are even not in a correct web-link format.

Response: All the links in the text have been checked and corrected. 

Comment No.14: part a and part b should be merged in a single figure as Fig. 2

Response: The modifications have been considered.

Comment No.15: if possible, a better image could be used

Response: The authors have used their best image. So Fig 2 only was merged into a single image and the resolution was improved in his section. 

Comment No.16: italic

Response: The modifications have been considered. 

Comment No.17: the frame on this motif should be replaced or the letters can be differently colored to emphasize

Response: According to the reviewer's comment, the frame was replaced with the ▼ symbol above the motif letters, in this figure. A few errors in the typography of motif HXXXD were also corrected inside the text.

Comment No.18: lowercase

Response: The modifications have been considered

Comment No.19: Please correct the software name which should be PredictNLS

Response: Thanks to the reviewer for the comment. The name of the software was correct with PredictNLS.

Comment No.20: make sure this figure is legible

Response: The author attempted to replace the figure with a more legible one.

Comment No.21: part a and b should be merged into a single figure

Response: Thanks to the reviewer for the comment. Two separate figures, which are the outputs of the software, were designed into one figure. According to similar references (e.g., Safavi et al., 2019; Elengoe et al., 2014; pandey et al., 2014), parts A and B do not merge into a single image. So, two parts are labeled A and B in this figure, but if the respected reviewer suggests it, the author will merge the two parts in the next revision.

- Safavi, A., Kefayat, A., Sotoodehnejadnematalahi, F. et al. In Silico Analysis of Synaptonemal Complex Protein 1 (SYCP1) and Acrosin Binding Protein (ACRBP) Antigens to Design Novel Multiepitope Peptide Cancer Vaccine Against Breast Cancer. Int J Pept Res Ther 25, 1343–1359 (2019). https://doi.org/10.1007/s10989-018-9780-z

-Elengoe, A.; Naser, M.A.; Hamdan, S. Modeling and Docking Studies on Novel Mutants (K71L and T204V) of the ATPase Domain of Human Heat Shock 70 kDa Protein 1. Int. J. Mol. Sci. 2014, 15, 6797-6814. https://doi.org/10.3390/ijms15046797

-Pandey, Saurabh & Negi, Yogesh & Reddy, Subramanyam & S, Krishna Murthy & Arora, Sandeep & Kaul, Dr. Tanushri. (2014). Modeling and phylogenetic analysis of cytosolic ascorbate peroxidase (OsAPX1) from rice reveal signature motifs that may play a role in stress tolerance. Bioinformation. 10. 119-123. 10.6026/97320630010119. 

Discussion:

Comment No.22: discussion should be improved. the most parts look like a repetition of the results section. the importance and potentials of the findings/results should be interpreted and elaborated. the significance of elucidation of the sequence and the structure of this enzyme should be clearly indicated. the recent studies about this enzyme (characterization or heterologous expression in microbial hosts) should be mentioned here and the potential of this study should be discussed. 

please see below for other notes.

Response: Thanks to the reviewer for the suggestion. The author has been tried to improve the discussion part by adding some new complementary references and rewriting some sentences and also, adding some new sentences to make more explanations and clarifications. 

Comment No.23: in contrast,

Response: The modification “In contrast” has been done.

Comment No.24: should be reworded or removed

Response: the sentence “this process was the biological process in the pc biosynthesis” was removed.

Comment No.25: a single aa cannot be an active site. rather, it should be in active site

Response: Thanks for the reviewer comment. Based on the suggestion, the authors changed the sentences and, as a result, the subsequent related sentences. the modified sentences: “Therefore, His-164 is found in the active site of the T5AT protein (C. avellana). The active sites in the T5AT in T. cuspidate were also included His-164 and Asp-373. As a result, it is most likely that Asp-381 is another amino acid in the active site involved in the T5AT (C. avellana).”

Comment No.26: activity site is not a correct term. should be active site. this term should be corrected in all parts of the text

Response: The “activity site” has been replaced with the “active site” in the whole text. 

Comment No.27: this part should be improved too

Response: The modification was made according to the suggestion. 

Comment No.28: conclusion is poorly written, should be improved

Response: The author tried to improve the conclusion by replacing and rewriting some sentences.

Comment No.29: not a clear statement, should be improved/clarified

Response: The sentence “It can be deduced that the taxadien-5α-ol-O-acetyltransferase enzyme has undergone many changes during development.” was replaced with “It can be assumed that the taxadien-5α-ol-O-acetyltransferase enzyme has undergone numerous changes during its evolution”.

Comment No.30: This gene was in the primary PC biosynthetic pathway. 

Response: the mentioned sentence was removed. 

Comment No.27: Please correct the underlined parts in fig 1

Response: Fig 1 was modified according to the suggestion.

Response to Comments by Reviewer# 2

We would like to thank the Reviewer for his/her constructive comments and suggestions. Based on his/her remarks, various revisions have been made to improve the quality of the manuscript. Our responses to the review comments are presented next.

Abstract

Comment No.1: Line no. 16 – 18: ‘and tissue cultures of Taxus 16 species and’ not correlating here. Reframe the sentence.

Response: Thanks for the reviewer comment. According to the suggestion, the authors improved the sentence with “Natural materials including tissue cultures of Taxus species and, most recently, hazelnut (Corylus avellana L.) represent the most promising sources of this compound.”

Comment No.2. Line no. 20: change as ‘this study was undertaken to …..

Response: The sentence “this study was undertaken to…” has been replaced with the “this paper aimed to identify taxadiene-5α-ol-O-acetyltransferase (TDAT) in hazelnut.” 

Introduction

Comment No.3: Line no. 44 – 46: Sentence starting as ‘In particular……’ rewrite the sentence

Response: the mentioned sentence was improved as follows: “Particularly, hazelnut (Corylus avellana L.) [6], some microorganisms like yew endophytic fungi, and hazelnut endophytic fungi [7, 8] are the other raw resources of the PC production.”

Comment No.4: Line no. 55 – 58: Sentence starting as ‘The deep and detailed understanding’ rewrite the sentence.

Response: The mentioned sentence was replaced with “A comprehensive understanding of the PC biosynthetic pathway in hazelnut, particularly the genes encoding rate-limiting enzymes and the enzymes catalyzing these reactions, are influential factors for developments in metabolic engineering and and for the production of PC to be significantly increased”

Comment No.5: Line no. 65: Change as ‘The initial main precursor of PC’

Response: The sentence was modified according to the comment.

Materials and methods

Comment 6: Line no. 88: ‘were procured from Merck’

Response: The modification was considered

Comment 7: Line no. 89: ‘were procured from Merck’

Response: The modification was considered

Comment 8: Line no 94: dissolving

Response: The modification was considered

Results

Comment 9: Line no. 195: (non-elicited)

Response: The modification was considered

Comment 10: Line no. 196 and 197: Remove (0)

Response: The modification was considered

Comment 11: Line no. 200: change as ‘for 1423 bp and were subjected to sequencing.

Response: The modification was considered

Comment 12: Line mo. 305 – 308: consider rewriting the sentence

Response: The sentence was modified as follows: “Primarily, similar and most close sequences to the T5AT sequence were obtained using the protein BLAST database. The hydroxycinnamoyl-COA shikimate/quinate hydroxycinnamoyl transferase enzyme, which it’s 3D-structure (PDB code; 4G0B) was identified by Lallemand et al. [44], was considered as a template for the T5AT protein.”

General comment:

Comment 13: The authors have to improve the language in the entire manuscript to meet the journal standard.

Response: Thank you very much to the reviewer. The author attempted to proofread and refine the entire manuscript's grammar to meet the journal’s standard. The grammatical revisions have been marked with track changes inside the text.

Best Regards

 

6- PLOS authors have the option to publish the peer review history of their article (what does this mean?). If published, this will include your full peer review and any attached files.

Response: No.

---

## [Decision Letter · Decision Letter 1]

25 Jun 2021

PONE-D-21-05459R1

Identification and in-silico characterization of taxadien-5α-ol-O-acetyltransferase (TDAT) gene in Corylus avellana L.

PLOS ONE

Dear Dr. Rahpeyma,

Thank you for submitting your manuscript to PLOS ONE. After careful consideration, we feel that it has merit but does not fully meet PLOS ONE’s publication criteria as it currently stands. Therefore, we invite you to submit a revised version of the manuscript that addresses the points raised during the review process.

We look forward to receiving your revised manuscript.

Kind regards,

Balamurugan Srinivasan

Academic Editor

PLOS ONE

Reviewers' comments:

Reviewer's Responses to Questions

**Comments to the Author**

1. If the authors have adequately addressed your comments raised in a previous round of review and you feel that this manuscript is now acceptable for publication, you may indicate that here to bypass the “Comments to the Author” section, enter your conflict of interest statement in the “Confidential to Editor” section, and submit your "Accept" recommendation.

Reviewer #2: All comments have been addressed

Reviewer #3: (No Response)

2. Is the manuscript technically sound, and do the data support the conclusions?

Reviewer #2: Yes

Reviewer #3: Yes

3. Has the statistical analysis been performed appropriately and rigorously? 

Reviewer #2: Yes

Reviewer #3: N/A

4. Have the authors made all data underlying the findings in their manuscript fully available?

Reviewer #2: Yes

Reviewer #3: (No Response)

5. Is the manuscript presented in an intelligible fashion and written in standard English?

Reviewer #2: Yes

Reviewer #3: (No Response)

6. Review Comments to the Author

Reviewer #2: (No Response)

Reviewer #3: The submitted manuscript by Shirazi et al., shows the identification of TDAT gene and provides interesting in silico analyses of the TDAT gene. The data will be interesting to the readership; specifically, the community works on PC biosynthetic pathway. However, the paper needs revision, particularly in terms of gene identification and the methodology adopted for the study.

The authors shall rephrase all the RT-PCR into sqRT-PCR, which misinterpret that the authors had performed real-time quantitative PCR. Based on figure 2, all lanes 2,3 and 4 showed the smear, and, intriguingly, only the 150 uM resulted in the amplification of the TDAT gene.

What is the dark line present in figure 2 between lane 3 and lane 4? The authors claimed that gel cropped from different gels was differentiated by white lines; however, no such differentiation exists in this figure.

The authors claimed that they had purified the PCR product obtained from lane 5. Was it gel elution or purification? P7 L155- authors mentioned as gel extraction- these details may be precisely given in the results section.

Moreover, lane 7 does not seem to contain the purified products, instead, it looks more smear than the unpurified products (Lane 5). The authors explain this context.

Based on figure 2, the presence of TDAT amplicon in the lanes 2, 3 an 4 can not be ruled out. There might be some problem with the purification of PCR amplicons from the gel.

Collectively, to avoid all these misperceptions, the authors are highly recommended to perform real-time quantitative PCR analysis to detect the relative mRNA level of TDAT gene at 0, 50, 100 and 150 uM of MeJa.

The number of figures may be reduced. Few figures may be shifted to the Supplementary file. For instance, Figs 1, 2, 3, 5, 7 may be moved to supplementary file.

Minor points:

P4 L96: What is the content/quantity of callus taken to generate suspension culture as per the given protocol, which may be provided in the methods section.

P4 L100: Which term the authors imply: produced or procured?

P6 L150: µm

P6 L 154: Primer sequences may be given in the table

P19 L454-455: This claim warrants the use of qPCR analysis, considering the inconsistent result of the given sqRTPCR gel pictures.

7. PLOS authors have the option to publish the peer review history of their article (what does this mean?). If published, this will include your full peer review and any attached files.

Reviewer #2: **Yes: **Muthukrishnan Arun

Reviewer #3: No

---

## [Author Response · Author response to Decision Letter 1]

7 Aug 2021

Dear Editor-in-Chief,

We would like to deeply thank the editorial office and the reviewers for the constructive comments which helped us to improve the paper and thank you for giving us the opportunity to submit a revised draft of our manuscript titled "Identification and in-silico Characterization of taxadien-5α-ol-O-acetyltransferase (TDAT) Gene in Corylus avellana L". The authors tried to address all reviewers’ comments in the revised manuscript. We hope our revision has improved the paper to a level of their satisfaction. The revisions in the manuscript have been marked highlighted with track changes in a file labeled ‘Revised manuscript with track changes.

Enclosed, please find the revised version of the manuscript entitled: 

“Identification and in-silico Characterization of taxadien-5α-ol-O-acetyltransferase (TDAT) Gene in Corylus avellana L.”

By:

Mona Raeispour Shirazi, Sara Alsadat Rahpeyma, Sajad Rashidi Monfared, Jafar Zolala and Azadeh Lohrasbi-Nejad1

Sincerely yours,

Dr. Sara Alsadat Rahpeyma

Assistant Professor of Plant Breeding

Department of Agricultural Biotechnology

Faculty of Agriculture

Shahid Bahonar University of Kerman

Iran 

Response to Comments by Reviewer 3

We'd like to express our gratitude to the Reviewer for his or her constructive comments and suggestions. Based on his/her remarks, various revisions were made to improve the quality of the manuscript. The modifications were highlighted with track changes through the revised manuscript. The next sections contain our responses to the dear reviewer comments. 

Comment No.1: The authors shall rephrase all the RT-PCR into sqRT-PCR, which misinterpret that the authors had performed real-time quantitative PCR.

Response: Much appreciated to the reviewer for the kind comments on the manuscript. In this study, the novel EST assembly strategy has been applied for identification of TDAT gene, and subsequently reverse transcription (RT)-PCR technique was developed to isolate TDAT gene from cDNA. As you know, the term “RT-PCR” is used for amplification, and semi-quantitative reverse transcription (sqRT-PCR) is a simple and specific method for relative quantitative RNA and amplification and determination of the quantity of PCR products. Since in this paper, the quantity measure was not intended, so it seems that RT-PCR is a more reasonable term in this case. It’s also worth to noting that the term “RT-PCR” was use in different studies (e.g. - Molecular cloning and expression analysis of glutathione reductase gene in Chlamydomonas sp. ICE-L from Antarctica, Marine Genomics, Volume 5, 2012. - Dongmei Yin, Dian Ni, Lili Song, Zhiguo Zhang, Isolation of an alcohol dehydrogenase cDNA from and characterization of its expression in chrysanthemum under waterlogging, Plant Science, Volume 212, 2013. - Chun-Hong Li, Yong-Qing Zhu, Yu-Ling Meng, Jia-Wei Wang, Ke-Xiang Xu, Tian-Zhen Zhang, Xiao-Ya Chen, Isolation of genes preferentially expressed in cotton fibers by cDNA filter arrays and RT-PCR, Plant Science, Volume 163, Issue 6, 2002. - Iwao, Kyoko, Watanabe, Takashi, Fujiwara, Yoshiyuki, Takami, Koji, Kodama, Ken, Higashiyama, Masahiko, Yokouchi, Hideki, Ozaki, Kouichi, Monden, Morito, Tanigami, Akira. Isolation of a novel human lung-specific gene, LUNX, a potential molecular marker for detection of micrometastasis in non-small-cell lung cancer. International Journal of Cancer. Int. J. Cancer)

Comment No.2: Based on figure 2, all lanes 2, 3 and 4 showed the smear, and, intriguingly, only the 150 uM resulted in the amplification of the TDAT gene. 

Response: As you know, the smear is commonly seen in the background of agarose gel electrophoresis of RT-PCR products. Moreover, In the case of this study, it was very difficult to obtain RNA samples with high concentrations of total RNA from cultured cells of hazelnut. For this reason, we had to use maximum amounts of total RNA samples and RT products in RT and subsequent PCR reactions. These conditions were the main reasons for the observation of smear in the background of agarose gel. The authors also, carried out this experiment aimed at significant expressing of the TDAT gene, several times and it was confirmed from the results that the expression was not detected at low concentrations of elicitors. About the amplification of desired PCR products only in reactions with RNA from 150 µM MeJA, Nevertheless, according to the dear reviewer comment, and to avoid of the claims related with comparative experiments, the phrase ‘while TDAT expression was induced at only 150 µM of MeJA’ was revised with the phrase ‘and it was perceivable at higher concentration of MeJA’(P8 line219 and 220), and the phrase “of an exact concentration” was deleted from P19-line 507 

Comment No.3: What is the dark line present in figure 2 between lane 3 and lane 4? The authors claimed that gel cropped from different gels was differentiated by white lines; however, no such differentiation exists in this figure. 

Response: That's correct, and thanks for bringing it out. The trimmed areas of figure 2 had been distinguished in the original figure (S1_ row_ images.pdf) attached to the supplementary information. So the change was considered, and the dark line was replaced with white lines. During the previous edition, the figure 2 was also modified in response to feedback from reviewers.

Comment No.4: The authors claimed that they had purified the PCR product obtained from lane 5. Was it gel elution or purification?.

Response: It was gel purification using Ron’s Gel Extraction Kit (BIGRON, cat no.: 802501) for 1423 bp. It is mentioned at page 7 line 155.

Comment No.5: P7 L155-authors mentioned as gel extraction- these details may be precisely given in the results section..

Response: Many thanks for the comment. The phrase “using Ron’s Gel Extraction Kit (BIGRON, cat no.: 802501)” was deleted from the results.

Comment No.6: Moreover, lane 7 does not seem to contain the purified products, instead, it looks more smear than the unpurified products (Lane 5). The authors explain this context.

Response: Many thanks for these precise comments. As previously stated, the 1423 bp amplified DNA fragment was purified from an agarose gel using Ron’s Gel Extraction Kit (BIGRON, cat no.: 802501). Despite all the optimizations, the target PCR fragment was so thin that we had to use a large gel slice in the purification process. The smear in the background of Lane 7 is caused by utilizing the maximum amount of an agarose gel in the purification process. It's also worth noting that we designed PCR primers using the TDAT assembly sequence derived from an EST library as a template. To amplify the complete CDS in PCR reactions, we had to locate the primers at the end of the target sequence. As a result, we faced significant challenges in detecting and overcoming false priming and other qualitative criteria in designed primers. Undesirable bands have been amplified in PCR reaction possibly due to the false priming of PCR primers. However, we obtained a clean PCR product when using the purified PCR product as the template in nested-PCR reactions (S3 Fig, page 9).

Comment No.7: Based on figure 2, the presence of TDAT amplicon in the lanes 2, 3 an 4 can not be ruled out. There might be some problem with the purification of PCR amplicons from the gel. Collectively, to avoid all these misperceptions, the authors are highly recommended to perform real-time quantitative PCR analysis to detect the relative mRNA level of TDAT gene at 0, 50, 100 and 150 uM of MeJa.

Response: Thank you so much for your detailed comment. Certainly, qRT-PCR is most reliable choice for comparative expression studies of TDAT gene. However, as stated in the title, the major goal of this study was identification and sequence analysis of the target gene in hazelnut. Therefore, we focused our efforts on providing a detectable source of TDAT mRNA for use in gene isolation assays. We performed many PCR reactions on different amounts of RT samples from different cell samples. Every time, we found the target mRNA only in cells induced with 150 µM MeJA. With this reason, the work is only based on common RT-PCR reactions not semi quantitative or quantitative ones. It was enough to our purpose of the study. We checked throughout the manuscript and made essential revisions to ensure there are no claims related with comparative experiments. Please refer to P8 line 219 and 220; P19-line 507 in the revised manuscript. 

Comment No.8: The number of figures may be reduced. Few figures may be shifted to the Supplementary file. For instance, Figs 1, 2, 3, 5, 7

Response: The modification was considered and the fig 1, 2, 5 and 7 were shifted to the supplementary file.

Comment No.9: P4 L96: What is the content/quantity of callus taken to generate suspension culture as per the given protocol, which may be provided in the methods section.

Response: Almost 1 g white and friable callus was taken and suspended in 30 mL liquid media to generate suspension culture. The modification was added to the text.

Comment No.10: Which term the authors imply: produced or procured?

Response: The term produced was implied. It was recommended with the other reviewer.

Comment No.11: µm

Response: Thanks to the reviewer for the comment. The modifications have been considered.

Comment No.12: Primer sequences may be given in the table

Response: The primers’ sequence was added in S1 table.

Comment No.13: P19 L454-455: This claim warrants the use of qPCR analysis, considering the inconsistent result of the given sqRT-PCR gel pictures.

Response: Thanks for these precise comments from the reviewer. The authors did not do quantitative analysis because their efforts were focused on obtaining a reliable and detectable supply of TDAT mRNA for use in gene isolation assays, as stated in earlier comments. Using different concentrations of elicitors, in this paper, indicates valuable results of using elicitation for identification of low-expressed taxol genes in hazelnut or even other genes. Please consider our more explanations under the reviewer suggestions about qPCR.

---

## [Decision Letter · Decision Letter 2]

13 Aug 2021

Identification and in-silico characterization of taxadien-5α-ol-O-acetyltransferase (TDAT) gene in Corylus avellana L.

PONE-D-21-05459R2

Dear Dr. Rahpeyma,

We’re pleased to inform you that your manuscript has been judged scientifically suitable for publication and will be formally accepted for publication once it meets all outstanding technical requirements.

Kind regards,

Balamurugan Srinivasan

Academic Editor

PLOS ONE

Additional Editor Comments (optional):

Reviewers' comments:

Reviewer's Responses to Questions

**Comments to the Author**

1. If the authors have adequately addressed your comments raised in a previous round of review and you feel that this manuscript is now acceptable for publication, you may indicate that here to bypass the “Comments to the Author” section, enter your conflict of interest statement in the “Confidential to Editor” section, and submit your "Accept" recommendation.

Reviewer #3: (No Response)

2. Is the manuscript technically sound, and do the data support the conclusions?

Reviewer #3: (No Response)

3. Has the statistical analysis been performed appropriately and rigorously? 

Reviewer #3: (No Response)

4. Have the authors made all data underlying the findings in their manuscript fully available?

Reviewer #3: (No Response)

5. Is the manuscript presented in an intelligible fashion and written in standard English?

Reviewer #3: (No Response)

6. Review Comments to the Author

Reviewer #3: Since the authors clarified all the queries raised by this reviewer and also considering the revisions accorded in the submitted version, the manuscript in its present form may be accepted for publication.

7. PLOS authors have the option to publish the peer review history of their article (what does this mean?). If published, this will include your full peer review and any attached files.

Reviewer #3: No

---

## [Editor Report · Acceptance letter]

19 Aug 2021

PONE-D-21-05459R2 

Identification and *in-silico* characterization of *taxadien-5α-ol-O-acetyltransferase* (TDAT) gene in *Corylus avellana* L. 

Dear Dr. Rahpeyma:

I'm pleased to inform you that your manuscript has been deemed suitable for publication in PLOS ONE. Congratulations! Your manuscript is now with our production department. 

Kind regards, 

on behalf of

Dr. Balamurugan Srinivasan 

Academic Editor

PLOS ONE